# Ku-Band Mixers Based on Random-Oriented Carbon Nanotube Films

**DOI:** 10.3390/nano14050450

**Published:** 2024-02-29

**Authors:** Mengnan Chang, Jiale Qian, Zhaohui Li, Xiaohan Cheng, Ying Wang, Ling Fan, Juexian Cao, Li Ding

**Affiliations:** 1Key Laboratory of Luminescence & Optical Information, Ministry of Education, School of Physical Science and Engineering, Beijing Jiaotong University, Beijing 100044, China15033101218@163.com (Z.L.); lfan@bjtu.edu.cn (L.F.); 2Hunan Institute of Advanced Sensing and Information Technology, Xiangtan University, Xiangtan 411105, China; jiale_qian@163.com (J.Q.); jxcao@xtu.edu.cn (J.C.); 3Academy for Advanced Interdisciplinary Studies, Peking University, Beijing 100871, China; 18811329877@163.com; 4Key Laboratory for the Physics and Chemistry of Nanodevices and Center for Carbon-Based Electronics, School of Electronics, Peking University, Beijing 100871, China

**Keywords:** carbon nanotubes, radio frequency transistor, mixer, conversion gain, linearity

## Abstract

Carbon nanotubes (CNTs) are a type of nanomaterial that have excellent electrical properties such as high carrier mobility, high saturation velocity, and small inherent capacitance, showing great promise in radio frequency (RF) applications. Decades of development have been made mainly on cut-off frequency and amplification; however, frequency conversion for RF transceivers, such as CNT-based mixers, has been rarely reported. In this work, based on randomly oriented carbon nanotube films, we focused on exploring the frequency conversion capability of CNT-based RF mixers. CNT-based RF transistors were designed and fabricated with a gate length of 50 nm and gate width of 100 μm to obtain nearly 30 mA of total current and 34 mS of transconductance. The Champion RF transistor has demonstrated cut-off frequencies of 78 GHz and 60 GHz for *f_T_* and *f_max_*, respectively. CNT-based mixers achieve high conversion gain from −11.4 dB to −17.5 dB at 10 to 15 GHz in the X and Ku bands. Additionally, linearity is achieved with an input third intercept (IIP3) of 18 dBm. It is worth noting that the results from this work have no matching technology or tuning instrument assistance, which lay the foundations for the application of Ku band transceivers integrated with CNT amplifiers.

## 1. Introduction

Semiconducting carbon nanotubes (CNTs) [1] possess extremely high carrier mobility (up to 100,000 cm^2^/(V·s) at room temperature) and saturation velocity (over 10^8^ cm/s), as well as small intrinsic capacitance [2]. The remarkable carrier mobility and saturation velocity of carbon nanotubes make them highly suitable for high-speed and high-frequency applications in radio frequency (RF) electronics [3]. This intrinsic advantage of carbon nanotubes paves the way for the development of carbon-based RF devices and circuits [4,5,6]. Carbon–carbon bonds [7], on the other hand, are the strongest chemical bonds in nature and have extremely high thermal conductivity [8], thus guaranteeing the stability of CNT-based devices when operating in RF circuits. Carbon-based RF devices have been extensively studied and are theoretically expected to reach terahertz in short channels [9,10]. However, limited by the lack of purity and large-scale uniformity of early carbon nanotube materials, the actual performance of carbon nanotube RF transistors is far worse than expected. In recent years, with improvements in semiconductor CNT purity and the optimization of device structure, carbon nanotube RF transistors [11,12] and integrated circuits [13,14,15,16] have made good progress. And in the past two years, the cut-off frequency of carbon nanotube-based radio frequency transistors (RF FETs) has increased from 100 GHz [11,17] to more than 300 GHz [18,19] in the terahertz field, making important breakthroughs. Randomly oriented carbon nanotube film materials are often considered to be limited to medium- and low-performance devices and circuits. The large-scale uniformity of randomly oriented carbon nanotube films and the emergence of quasi-parallel arrangements in ultra-short channels (with significantly reduced overlap between carbon nanotubes) enable the construction of high-speed RF devices and circuits based on randomly oriented carbon nanotube films. High-performance transistors and ICs built by randomly oriented carbon nanotube films have been realized, such as ring oscillators (ROs) with 5.54 GHz of oscillation frequency [20] and RF transistors with over 100 GHz of pad de-embedding cut-off frequency [11]. Besides the potential in speed, randomly oriented carbon nanotubes have also been reported in works on preliminary RF devices such as amplifiers. For example, in 2021, Zhou et al. demonstrated RF amplifiers [21] with 11 dB of power gain and 15 dBm of output third intercept point (OIP3) in the X to Ku band for potential application in satellite communication.

CNT-based devices could be used for frequency conversion, including frequency multipliers or mixers applied in RF transceivers. Mixers are very common in communication systems, which use a local oscillation (LO) signal to transmit (receive) an input signal of a certain frequency through an up-conversion (down-conversion) operation. Mixers based on transistors generally work in two ways [22], namely, passive and active. To date, most of the reported carbon-based mixers (including CNT and graphene) have been passive resistance mixers. The double-gate Bernal stacked double-layer graphene transistor passive mixer reported by Tian et al. exhibited −12.7 dB conversion gain at 2 GHz [23]. Moon et al. reported that passive mixers based on graphene exhibit 27 dBm of IIP3 (input third intercept) and −14 dB conversion gain at 2 GHz [24]. Lyu et al. demonstrated a double-balanced gate-pumped passive mixer integrated circuit that exhibited 21 dBm IIP3 and −33 dB conversion gain at 3.5 GHz [25]. Compared to passive resistance mixers, there have been fewer reports on active transconductance mixers than graphene-based active mixers, reported by Lin et al. to exhibit −27 dB conversion gain at 4 GHz [26]. Che Y. et al. demonstrated a conversion gain of −24.5 dB at 1 GHz and an IIP3 of 22.3 dBm at 5 GHz using aligned carbon nanotube array-based devices [27]. It is worth noting that conversion gain and IIP3 are relatively low for carbon-based RF mixers in the sub-6 GHz band, so it is particularly important to continue to improve the frequency conversion capability of carbon-based RF mixers. Furthermore, randomly oriented carbon nanotube film-based RF devices have developed for several years with remarkable progress, while demonstrations of frequency conversion devices such as mixers with performances compatible with Ku band amplifiers are still lacking [21].

In this work, we designed and fabricated RF FETs based on randomly oriented carbon nanotube films and constructed an active mixer with the transistor as the core. The frequency conversion capability of the mixer in the X and Ku bands was explored. The wider gate width of the transistor allowed for lower internal resistance, which allowed for better and easier impedance matching with the test system. CNT-based RF transistors with an on-state current of 30 mA and peak transconductance of 34 mS achieved a cut-off frequency (*f_T_*) and maximum oscillation frequency (*f_max_*) of 78 and 60 GHz, respectively. Experiments and analyses demonstrated that CNT-based active mixers have better conversion gain of −11.4 dB~−17.5 dB at 10 GHz to 15 GHz for the X and Ku band and an IIP3 of about 18 dBm, which is compatible with a CNT-based RF amplifier in terms of process and performance. It is worth noting that, different from previous related works on the pursuit of limit RF performance, this work is a preliminary attempt to determine the actual performance of RF electronic devices without an on-chip matching network and tuning instrument assistance in order to promote the real performance of RF electronic devices based on randomly oriented carbon nanotubes. The CNT-based mixer used in this work is compatible with the CNT-based RF amplifier [21] and silicon-based CMOS, which heralds the potential for building practical RF transceiver systems in satellite communications and future system-on-chip integration on silicon.

## 2. Materials and Methods

### 2.1. Preparation of the Randomly Oriented Carbon Nanotube Films

#### 2.1.1. Preparation of the CNT Films

Randomly oriented carbon nanotube films were prepared using the method previously reported [28]. Raw arc-discharged single-walled carbon nanotubes (SWNTs) were purchased from Carbon Solution Inc. (CSI, Riverside, CA, USA). This powder was obtained by high-voltage arc discharge, so these carbon nanotubes were not purified by a semiconductor tube. The dispersant (poly[9-(1-octanoyl)-9H-carbazole-2,7-diyl] (PCz)) was synthesized via Suzuki polycondensation for dispersion. PCz is an excellent polymer material for the dispersion and purification of carbon nanotubes. The carbon tubes obtained after the dispersion of PCz have the advantages of few defects and high semiconductor purity. At the same time, PCz also has the radius (chirality) selection for carbon nanotubes with a diameter of about 1–2 nm [29]. First, 200 mg of PCz was mixed with 100 mg of CNTs by adding 100 mL of toluene. The mixed carbon nanotube solution was sonicated for 30 min using a cell grinder (Sonics VC500, Sonics & Materials, Inc., Newtown, CT, USA) with a 50% duty cycle. The solution was centrifuged at 30,000× *g* for 30 min to remove most of the multi-tubular bundles and insoluble material, and 90% of the top layer was removed, using a high-speed centrifuge model (Allegra X-64R, Beckman Coulter, Brea, CA, USA) for the entire process. This was centrifuged at 50,000× *g* for 2 h to remove trace amounts of metal nanotubes. The final supernatant was collected to obtain a high-purity solution of semiconductor carbon nanotubes. The dense carbon nanotube films were fabricated by the dip-coating method, and the high-resistivity SiO_2_/Si substrate was immersed in the collected supernatant for 48 h. After deposition, the substrate was blow-dried by high-purity nitrogen and baked in a tube furnace (Thermo Scientific Lindberg/Blue Moldatherm 1100 °C (Waltham, MA, USA)) for 30 min at 120 °C.

#### 2.1.2. Pretreatment of the CNT Films before Transistor Fabrication

Next, the yttrium oxide coating and decorating process was carried out [30]. First, 3 nm of yttrium metal (Y) was deposited on the CNT films by electron-beam evaporation (DE400) and then thermally oxidized at 250 °C in a hot plate for 0.5 h. The yttrium oxide layer was removed by immersion in a 1:20 volume solution of hydrochloric acid for 5 min (this ensured that there were no residual yttrium oxide residues on the carbon nanotube films). Finally, repeated rinses with deionized water and isopropanol were used to obtain cleaner CNT films. The main purpose of this process was to remove the excess polymer molecules wrapped around the CNTs.

Then, the annealing process was carried out [20]. Annealing allowed us to further remove the polymer residues from the CNTs. The substrate with the CNTs was placed in a tube furnace (Thermo Scientific Lindberg/Blue Moldatherm 1100 °C), and the excess air was vented from the tube by passing 1000 sccm of argon gas. The annealing temperature was 600 °C, and the flow rates of argon and hydrogen were 300 sccm and 50 sccm, respectively, for 1 h.

#### 2.1.3. Characterization of the CNT Films

CNTs with semiconducting purities greater than 99.99% were deposited onto a SiO_2_/Si substrate to create randomly oriented CNT films with a relatively high level of uniformity. Figure 1a displays SEM images of these films after undergoing an Y_2_O_3_-based cleaning process and thermal annealing treatment.

### 2.2. Fabrication of Carbon Nanotube RF Transistors

The fabrication of RF transistors is a process based on a lift-off process. In this process, patterning techniques such as electron-beam lithography (Raith voyager, Dortmund, Germany), thin film deposition techniques such as electron-beam deposition (DE400) and atomic-layer deposition (Ultratech Cambridge Savannah G2, Waltham, MA, USA), etching equipment (Trion Minilock-Phantom III, Miami, FL, USA), and an optical microscope (Carl Zeiss Axio Imager, Göttingen, Germany) are mainly used. Here, the fabrication of two-finger as well as multi-finger top-gate carbon nanotube transistors with an air-spacer between the source and drain took place. First, the source-drain contacts (Pd/Au = 20/10 nm) were deposited by electron-beam evaporation on pretreated CNT films. The channel region was then patterned by electron beam lithography followed by oxygen plasma etching. A Ti/Au = 5/90 nm stacked film was deposited as the lower interconnection line. The overlap of the interconnects was covered by overexposed PMMA 200 K bridges as jumpers. HfO_2_ films with a thickness of 5 nm were then grown by ALD at 90 °C (40 cycles) as oxide dielectrics. A Ti/Pd/Au (0.3/5/180 nm) stacked film was deposited in the center of the channel as the gate. Finally, a Ti/Au = 10/400 nm metal film was deposited in the area to be connected as the gate electrode, the top metal, and the GSG structure. The details of the fabrication process are shown in Appendix A. The as-fabricated two-finger-configuration RF transistor based on randomly oriented carbon nanotube films is shown in Figure 1b of the SEM picture.

### 2.3. Measurement of S Parameter

As RF devices tend to operate in the high-frequency band, if a normal probe station is used for testing, the connection line and the measuring equipment will incur significant power losses in the high-frequency band. Therefore, a special high-frequency probe station (Cascade Summit 1100, Cascade Microtech, Beaverton, OR, USA), coaxial cable, semiconductor analyzer (Keithley 4200 (Keithley Instruments, Cleveland, OH, USA) and Agilent B1500 (Agilent Technologies, Santa Clara, CA, USA)), GSG probe, and vector network analyzer (Agilent E8363B) were used. At the same time, during the process of RF transistor fabrication, it should be noted that the spacing of pads should meet the specification of the 150 μm GSG probe. Before measurements, GSG probe heads and coaxial cables were standardized using off-wafer short-open-load-through (SOLT). DC bias *V*_gs_ and *V*_ds_ of CNT-based RF transistors were provided by semiconductor analyzers. The gate and drain were port “1” and port “2”, respectively. The vector network analyzer was used to measure the S-parameters of the two ports (S_11_, S_21_, S_12_, S_22_).

### 2.4. Measurement of Mixer

#### 2.4.1. Mixing Performance Measurement of the Mixer

In this work, the active mixer was constructed based on a carbon nanotube field-effect transistor (FET) as the core. Mixer measurement requires a variety of instruments, including a probe table (MPI TS200-THz, MPI Corporation, San Jose, CA, USA), power amplifier (TVTA 1840-10, East Setauket, NY, USA), vector network analyzer (Keysight N5247A, Santa Rosa, CA, USA), semiconductor analyzer, bias Tee (Keysight 11612BK01), and spectrum analyzer (Keysight) N9020B). The DC bias source (Keysight B2902A) was used to supply the power voltage to the CNT-based RF mixer. Before testing, SOLT calibration of the entire test route was required to compensate for power loss at high-frequency operation, resulting in more accurate mixer performance. The RF and local oscillator (LO) signals were combined into one signal source through a combiner (the combiner was reversed as a divider) to the gate as the input port and the drain as the output port. The intermediate frequency (IF) signal as output was obtained through the spectrometer. The conversion gain (CG) can be calculated by the following equation:(1)$$CG=10\log\frac{P_{IF}}{P_{RF}}$$
where *P*_IF_ is the output IF signal power, and *P*_RF_ is the input RF signal power.

#### 2.4.2. Linearity Measurement of Mixer

The 1 dB compression point (P1dB) is defined as the linear gain deviation of the mixer IF signal power as the RF input signal power increases. When the RF signal power increases to a certain level, the IF signal power no longer increases as the linear gain, and the IF signal power deviates from the RF signal power corresponding to the linear change. Theoretically, as the input RF signal power level increases, the output IF power level of third-order intermodulation increases three times faster than that of the fundamental. The third-order intermodulation point (IP3) is the intersection of the fundamental signal in an ideal linear system when the amplitude of the output signal is equal to the third-order intermodulation component, namely IP3, where the power of the third-order intermodulation is equal to the extrapolated value of the fundamental. The input and output power of the IP3 corresponds to IIP3 and OIP3, respectively. In this work, we measured the fundamental and third-order intermodulation scatter diagrams at 1:1 and 3:1, respectively. The intersection was extrapolated to obtain IP3.

## 3. Results and Discussion

### 3.1. Electrical Characterization of RF Transistors

Top-gated RF transistors were designed based on randomly oriented carbon nanotube films (~40–60 CNTs/μm, see Figure 1a) with the ground-signal-ground (GSG) test architecture for RF measurements and fabricated using the lift-off process described in the Materials and Methods section. Usually, metal stack Palladium and Gold (Pd/Au) of 20/10 nm were deposited onto CNT films to form the p-type contact. Atomic-layer deposition-grown 5 nm HfO_2_ is the high-k gate dielectric layer for efficient gate control. The wire connection was formed by the deposition metal stack Titanium/Gold (Ti/Au) with a thickness of 5/90 nm. In the overlapping region, a bridge of PMMA resist was overexposed and sandwiched between two metal layers to reduce the parasitic capacitance compared with high-k dielectric HfO_2_. First, we fabricated two-finger-configuration RF transistors to verify the manufacturing process and measurement method. The channel length (Lch) of the two-finger gate transistor was about 130 nm, the gate length (L_g_) was 50 nm, and the total gate width was 20 μm with one finger width of 10 μm. A high-magnification SEM image of the active region shows the two-finger configuration as the schematic shown in Appendix A. Our two-finger-configuration RF transistors exhibited a current density of nearly 0.4 mA μm^−1^ and peak transconductance (g_m_) of over 0.4 mS μm^−1^ at a bias (*V*_ds_) of −1.4 V (Appendix A).

To achieve high-performance RF transistors, we needed to design a large gate width [12,31] to ensure adequate drive capacity and, more importantly, to reduce the power loss caused by the resistance mismatch between the device and the 50 Ω impedance. It is worth noting that enlarging gate width for improving device current relies on material uniformity, which is more difficult for aligned CNT films, although they have much better normalized currents (namely, current density). So, we designed randomly oriented CNT film transistors with the twenty-finger configuration of 100 μm gate width (W_ch_) and 50 nm gate length (L_g_), as shown in Figure 2a. The total driving current (I_on_) reached 30 mA and peak transconductance (g_m_) reached 34 mS at a bias (*V*_ds_) of −1.2 V, presenting practical performance in randomly oriented CNT-based FETs (Figure 2b,c). The DC performance of carbon nanotube-based FETs can help us estimate the RF performance of transistors and also help us determine the DC bias conditions required for transistor RF measurements. As shown in Figure 2b, the total driving current of the CNT-based FET reached 30 mA, which met the requirements of RF measurement. Based on the DC performance of the fabricated 20-finger-configuration RF transistor, we characterized its cut-off frequency performance which is the basis for RF application. The RF performance of a transistor could be characterized by the current-gain cut-off frequency (*f_T_)*, defined as the frequency at which the current gain becomes unity, and the power-gain cut-off frequency (*f_max_*), defined as the frequency at which the power gain becomes unity. As shown in Figure 2d, the current gain (H_21_) and power gain (MSG/MAG) versus frequency curves demonstrated that the cut-off frequency (*f_T_*) and maximum oscillation frequency (*f_max_*) of the device reached 78 GHz and 60 GHz, respectively, both with the extraction slope of −20 dB/dec which met the minimum requirements for CNT-based RF transistors building RF devices operating in the Ku band (the working frequency is at least one-third of *f_max_*) [32].

### 3.2. Mixing Performance Characterization of Mixer

Next, we further explore the conversion capability of RF devices based on randomly oriented CNT films. Based on the developed randomly oriented CNTs RF FETs, we built active mixers for frequency conversion. Our active mixers were measured according to the measurement schematic diagram shown in Figure 3a; the RF signal (*f_RF_*) and the local oscillator signal (*f_LO_*) were combined and applied to the gate terminal which modulated the device to realize the mixing function. Two bias Tees were used before the gate and drain terminals, respectively, to combine DC and RF signals while reducing the interference between DC bias and RF signals. The output signal of the mixer, also denoted as IF signals, was obtained from the drain terminal. When measuring the mixer for the first time, in addition to calibrating the test line, it was necessary to clarify the basic conditions of mixer measurement, including bias conditions, local oscillator power, and operating frequency band, as shown in Appendix A. Figure 3b shows the typical mixing result of a mixer with two adjacent frequencies as input signals of an RF (radio frequency) signal (*f_RF_* = 10.1 GHz with power of −10 dBm) and an LO (local oscillator) signal (*f_LO_* = 10 GHz with power of 7 dBm) combined into the gate. The output signal of the intermediate frequency signal (*f_IF_*) was located right at 100 MHz, which was the frequency difference (*f_RF_* − *f_LO_* = 0.1 GHz). The typical conversion gain was about −11.4 dB, as shown in Figure 3c at 10 GHz of the X band.

According to S. A. Maas et al. [33], the conversion gain CG can be expressed by the transistor parameter as follows:(2)$$CG=\frac{g_{m,max}^{2}R_{L}}{16\omega_{1}^{2}C_{gs}^{2}R_{il}}$$
where *g*_m,max_ is the maximum transconductance, *R*_L_ is the load resistance of the IF signal port, ω_1_ is the RF signal frequency, *C*_gs_ is the capacitance between the gate and the source, and *R*_il_ is the resistance of the input loop, including the intrinsic resistance of the gate, source, and field effect transistor. From Equation (2), the transconductance of the RF transistor is vital for mixers achieving maximum conversion gain, which would decrease as frequency increases. Figure 3c shows the relationship between the input RF signal power (*P*_RF_) and output IF signal power (*P*_IF_) at three different frequencies (*f*_RF_) of 10 GHz, 12 GHz, and 15 GHz, covering from the X band to Ku band, while *P*_LO_ = 7 dBm. As the RF signal power scales up from −5 dBm to −0.5 dBm, the IF signal power correspondingly increases linearly without gain depression. The conversion gain is calculated by CG = *P*_IF_ − *P*_RF_, and it is −11.4 dB, −13.1 dB, and −17.5 dB, respectively, for three frequencies.

Furthermore, linearity is also needed for mixers and other RF devices to keep RF signals undistorted. We demonstrated the CNT-based mixer’s linearity by measurements shown in the Materials and Methods section. The 1 dB compression (P1dB) point for CNT-based mixers is demonstrated as 4 dBm in Appendix A. The third-order intercept point (IP3) is an important parameter to characterize the linearity of RF devices. In Figure 3d, we could extract the input third-order intercept point (IIP3) for our fabricated mixer, and the intermediate frequency signal (IF) power increased with the dependence of 10 dB/dec, while the third signal of intermodulation (IM) power followed the dependence of 30 dB/dec. By extrapolating IF and IM lines to intersect each other, we extracted 18 dBm of the input third-order intercept point (IIP3). The statistical distribution of CNT-based mixers is also shown in Appendix A.

### 3.3. Benchmarking of Mixers

We benchmarked the conversion gain and linearity of CNT-based mixers with previously reported results from nanomaterials [23,24,25,26,27,34] and conventional semiconductors [35,36], as shown in Figure 4, focusing on the X and Ku bands. The detailed data are summarized in Appendix A. From Figure 4a, we see that the conversion gain of our mixers is the best among all the nanomaterials such as graphene and MoS_2_ in the Ku band. As for IIP3, although previously reported carbon-based mixers have better performance than that in this work, the frequency band of CNTs is too low (less than 6 GHz) and the corresponding conversion gain of graphene is lower than ours. Compared to traditional semiconductors such as silicon and III-V compounds, our mixers still have a significant gap in conversion gain. In contrast, the IIP3 of carbon nanotube-based mixers has comparable performance to conventional semiconductors. It is worth noting that the performance achieved in this work has no on-chip matching network and tuning instrument assistance; our CNT-based mixer has a better frequency conversion capability in the Ku band. Therefore, considering the conversion gain and IIP3 parameters, the frequency conversion ability of CNT-based mixers demonstrates better performance than other nano-materials in the Ku band.

## 4. Conclusions

In summary, based on randomly oriented carbon nanotube films, we have designed and fabricated twenty-finger transistors with 50 nm gate length and 100 μm gate width. The design of the multi-finger structure can reduce the resistance of the device and reduce the power loss due to impedance mismatch. In particular, we investigated the frequency conversion capabilities of active mixers built on CNT FETs in the X and Ku bands. Thanks to the large-scale uniformity of the randomly oriented carbon nanotube film material in which the CNTs are approximately parallel under the small channel, the RF performance of our CNT FETs showed a cut-off frequency (*f_T_*) of 78 GHz and a maximum oscillation frequency (*f_max_*) of 60 GHz, respectively. The carbon nanotube-based mixers had conversion gains of −11.4 dB to −17.5 dB for input RF signals at 10 GHz to 15 GHz in the X and Ku bands and an IIP3 of 18 dBm. Without an on-chip matching network and tuning instruments assistance, our CNT-based mixer had a better frequency conversion capability in the Ku band compared to other nanomaterials such as graphene. These results demonstrate the performance advantages of CNT FETs and lay a foundation for the application of CNT-based mixers in the Ku band. Further, the mixers in this work are compatible with the carbon nanotube amplifiers in ref. [21] and silicon-based CMOS, which provide the possibility of Ku-band RF transceiver systems and more complex system-on-chip integration on silicon in the future.

## Figures and Tables

**Figure 1 nanomaterials-14-00450-f001:**
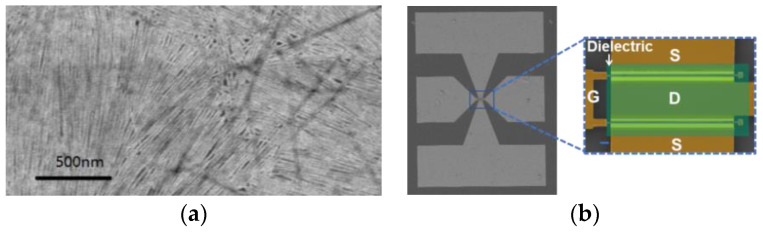
Carbon nanotube films and structures of RF transistors: (**a**) SEM image of randomly oriented carbon nanotube films; (**b**) SEM image of the whole two-finger CNT-based RF transistor with GSG electrode pads; the scale of the right illustration is 1 μm.

**Figure 2 nanomaterials-14-00450-f002:**
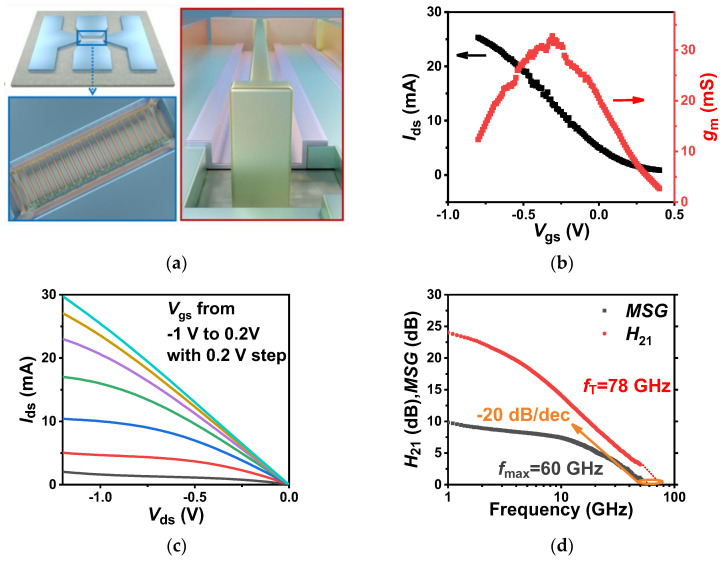
Structure of a twenty-finger-configuration RF transistor based on randomly oriented carbon nanotube films and its electrical characteristics: (**a**) schematic diagram of a twenty-finger CNT-based RF transistor at high-magnification SEM with a total channel width of 100 μm; (**b**) typical transfer characteristics of a 50 nm gate length twenty-finger RF transistor with *V*_ds_ at −1.2 V; (**c**) typical output characteristics of a 50 nm gate length twenty-finger RF transistor with *V*_gs_ from −1 V to 0.2 V with the step of 0.2 V; (**d**) frequency characteristics of a twenty-finger RF transistor with a measured cut-off frequency (*f*_T_) of 78 GHz and a maximum oscillation frequency (*f*_max_) of 60 GHz.

**Figure 3 nanomaterials-14-00450-f003:**
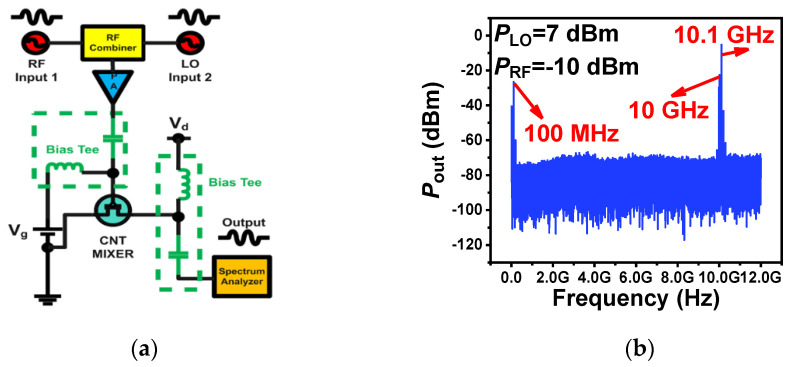
CNT-based active mixer: (**a**) measurement schematic diagram of the CNT-based active mixer; (**b**) the output spectrum of the CNT-based mixer at 10.1 GHz RF input signal with −10 dBm power and 10 GHz LO (local oscillator) signal with 7 dBm power; (**c**) the relationship between the RF signal power (*P*_RF_) and the intermediate frequency (IF) signal power (*P*_IF_) of the CNT-based active mixer at different operating frequencies from 10 GHz to 15 GHz; (**d**) 18 dBm IIP3 of the CNT-based active mixer operating at 10 GHz.

**Figure 4 nanomaterials-14-00450-f004:**
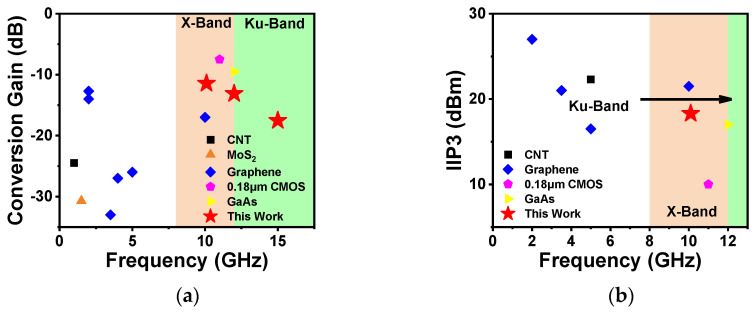
Benchmarking of mixers: comparison of (**a**) the conversion gain and (**b**) IIP3 of our CNT-based active mixers with those based on other nanomaterials [23,24,25,26,27,34] and conventional semiconductors [35,36].

## Data Availability

The data presented in this study are available on request from the corresponding author.

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
