# Peer review of "Ku-Band Mixers Based on Random-Oriented Carbon Nanotube Films"

_nanomaterials, 2024, doi:10.3390/nano14050450_

Round 1

Reviewer 1 Report

Comments and Suggestions for Authors

Dear authors,

The topic of the research of the manuscript could be interesting and it is very well related with the topic of the Nanomaterials journal.

In my opinion, the entire text of the manuscript must be re-written and re-organized and then, it can be resubmitted again for review.

The authors must clarify some details and data, which are missing. In order to improve this manuscript, I recommend some major changes and improvement as it is shown below.

1. The purpose and the main objectives of this research must be highlighted better in Abstract and in the end of Introduction section.

2. The Experimental section must be moved before to Results section. The authors should rename the Experimental section with “Materials tested and work method” in order to describe all materials tested and the equipment used in mechanical tests.

3. The authors must revise the details regarding the equipment used in order to show the following for each equipment: commercial name, type, manufacturer, main characteristics, testing parameters. Some pictures showing the specimens during the tests are also recommendable.

For example, the equipment used for the preparation referred in the following step is not given: Yttrium oxide coating and decoating process. First, 3 nm of yttrium metal (Y) was 287 deposited on the carbon nanotube films and then thermally oxidized at 250 °C for 0.5 288 hour.”

4. All sub-figures related with manufacturing and preparation of the specimens tested (like Fig. 1a, Fig. 1b, Fig. 1c, Fig. 2a, Fig. 3a) must be moved in the sub-sections of the chapter “Materials tested and work method”.

5. The SEM photos must show the zoom scale and the measure unit.

6. The authors should introduce a flow-chart to clarify the main steps covered in manufacturing and preparation of the specimens tested.

7. Description of the materials is too general. More details are required for CNs like the following: the commercial name, size (length and diameter), technical sheet of CNs should be cited and introduced in References list.

8. The quantities used in Eq. (2) are not explained.

9. Title of section  4.6 must be changed because the title must show the property measured by giving the entire name of that property. Just abbreviations are shown.

10. The Results section must be divided in sub-section related to the sub-sections of the chapter “Materials tested and work method”.

11. Legend of Fig. 3(c) must be moved over the graph.

12. A comparison of the results obtained for tested materials with the properties of “any other materials at Ku band” and “conventional semiconductors” must be introduced in “Results and discussions” section in order to correlate with the text from lines 103-108 in Introduction section and with the conclusion shown at lines 262-267 at the end of Conclusions section.

Comments on the Quality of English Language

Dear authors,

Please revise carefully the text of your manuscript and improve the expression in English.

Reviewer 2 Report

Comments and Suggestions for Authors

Despite the substantive content contained in the article, it is not editorially ready for publication. The article is not prepared in the format applicable at the publishing house.

The literature review requires significant improvement. The works include works from the last two years. This proves either that the work is of marginal importance (which is not this case) or that literary studies are insufficient. Additionally, the citations given in the first part are group citations. They need to be developed.

The descriptive part about the rules of conducting the experiment should be moved before the Results part.

The variables used in equation 2 should be described under the equation.

At the end of the Introduction part, the novelty of the paper should be better highlighted and the possible use of the obtained results should be provided in the summary.

Reviewer 3 Report

Comments and Suggestions for Authors

In their manuscript titled "Ku-band mixers based on random-oriented carbon nanotube films", Zhaohui Li and coworkers present carbon nanotube based devices with record performance within the Ku-band. These findings represent a significant leap forward in the field of RF-electronics using nanomaterials, and seem to have a large application potential for communication devices.

The data is presented in a clear and concise manner. The authors successfully compare their results with existing technologies, effectively highlighting the potential of their approach. The results are both significant and highly relevant to the field.

However, before considering this paper for publication, I suggest addressing the following points for clarification and improvement:

1. Figure 1a: A scale bar is missing for the SEM image.

2. Figure 2b: The authors should describe the image in more detail.

3. Figures 3c and d: Error bars are missing.

4. Figure 4: Error bars are missing.

Supporting Info:

Error bars are missing from the plots. The red lines in Figures S1 and S2 are not explained.

Given the suggestions provided, I recommend that the paper should be published after undergoing a minor revision.

Comments on the Quality of English Language

Except for a few minor errors, the quality of the English is very good.

Round 2

Reviewer 1 Report

Comments and Suggestions for Authors

Dear authors,

I read carefully the revised version of your manuscript and I observed that you made changes and improvements according to all comments from the first my review report. You added all details which were missing in the first version of your manuscript.

I don’t have another questions or comments regarding your research.

I may recommend the publishing of your manuscript in Nanomaterials journal.

Comments on the Quality of English Language

Dear authors,

Please revise carefully the text of your manuscript and improve the expression in English.

Reviewer 2 Report

Comments and Suggestions for Authors

The article has been improved compared to the last version and, in my opinion, is ready for publication